# The Molecular Basis of Pediatric Brain Tumors: A Review with Clinical Implications

**DOI:** 10.3390/cancers17091566

**Published:** 2025-05-04

**Authors:** Elias Antoniades, Nikolaos Keffes, Stamatia Vorri, Vassilios Tsitouras, Nikolaos Gkantsinikoudis, Parmenion Tsitsopoulos, John Magras

**Affiliations:** 1Second Department of Neurosurgery, Aristotle University School of Medicine, 546 36 Thessaloniki, Greece; nkeffes@yahoo.gr (N.K.); vktsitouras@gmail.com (V.T.); nikgkantsinikoudis@gmail.com (N.G.); par_tsits@yahoo.gr (P.T.); john.magras@gmail.com (J.M.); 2New York City Health and Hospital—Jacobi Medical Center Department of Pediatrics, Albert Einstein College of Medicine, Bronx, NY 10461, USA; vorri.stamatia@gmail.com

**Keywords:** mitogen-activated kinase pathway, phosphoinositide 3-kinase pathway, epigenetic regulation, sonic hedgehog pathway, wingless pathway, microRNA regulation, NFkB pathway, Hippo signaling

## Abstract

Central nervous system (CNS) tumors in the pediatric population are the second most common among other malignancies of this age spectrum. Carcinogenesis of these lesions is the result of the aberrant cell signaling process. The previous classification based on histologic characteristics cannot appropriately define the prognosis. According to the current classification, each tumor relates to specific identity mutations in one or more of these signaling steps. Lesions with deficient supervision of transcription, uncontrollable cell cycle initiation and apoptosis, or combined mutations are considered more aggressive. Consequently, treating physicians have to be acquainted with novel nomenclature. In this review, we review the recent literature and attempt to present the major tumor groups with their associated alterations.

## 1. Introduction

Central nervous system malignancies are the most frequent space-occupying lesions in children. They are second in incidence, only after leukemia in the etiological ranking. From an anatomical perspective, they can be divided into supratentorial and posterior fossa malignancies. Based on the time of diagnosis, these tumors are also categorized into congenital neoplasms (detected postnatally within the first two months postpartum), neoplasms of infancy (<1 year), and tumors of older children (>1 year) [1]. Their incidence is 5.4–5.6 cases per 100,000 individuals [2].

The current classification from the World Health Organization (WHO) highlights the molecular profile of these tumors. Specifically, the referred molecules may be present at either the cell signaling process, DNA transcription or throughout cell cycle [2].

Herewith, we performed a narrative review, aiming to present succinctly the new molecular identity of pediatric brain tumors. A novel nomenclature has entered the daily clinical praxis, which includes knowledge of cell signaling, growth, and duplication. We also intend to provide treating physicians with an easily approachable body of literature.

## 2. Oncologic Terms

The basic steps of carcinogenesis include non-ceasing proliferative signaling, evasiveness of tumor suppression, recalcitrance to apoptosis, novel angiogenesis, multiplicative deathlessness, infiltration, and dissemination [3].

Initiation is the establishment of a firm, mutated cell. This is an irreversible phenomenon. Initiation by itself does not appear to be sufficient for neoplastic growth. Only when these changes have not undergone repair and the cell completes its DNA synthesis are these aberrations considered constant [4].

The promotion phase entails the distinct proliferation of the initiated cell to shape a class of preneoplastic cells. A distinctive feature of the promotion is also the decrease in cell death. Tumor promoters do not induce either DNA mutations or tumors by themselves; rather, they act by sustaining proliferation of preneoplastic lesions [5]. Specifically, molecular alterations in tumor-suppressive genes or telomerase subunit alterations provoke uncontrolled cell proliferation and elongate this phase.

The progression stage includes enhanced DNA synthesis in the preneoplastic lesions, additional DNA aberrations, and translocations. Hence, independent proliferation is achieved [6].

Proliferation denotes cell number increment in a population due to exertion of the cell cycle and high cell mitosis [7]. The basic steps of the cell signaling sequence are schematically presented in Figure 1.

## 3. Neoplasm Categories

Gliomas are the most common tumors of childhood. They are further divided in low-grade and high-grade lesions [1], (Figure 1).

### 3.1. Low-Grade Gliomas

#### 3.1.1. Molecular Events

##### Alterations of Tyrosine Kinase Receptors

The attachment of a growth factor (GF) to a receptor unit ignites a signaling cascade. Receptor tyrosine kinases (RTKs) are highly specified cell surface receptors for polypeptide growth factors, hormones, and cytokines. Overall, 20 separate RTK groups exist [8]. Their structure comprises an outer membrane ligand-attaching segment, a single transmembrane helix, and a cytoplasmic segment, which contains a proximal-to-membrane modificatory area, a tyrosine kinase domain (TKD), and a carboxyl (C-) appendage [9] (Figure 2).

FGFR (Fibroblast Growth Factor receptor) Alterations

*FGFR1* alterations respond to three scenarios: *FGFR1* point alterations, *FGFR1-TACC1* conjunction, and *FGFR1-TKD* duplications. *FGFR1* alteration occurs in 5–10% of patients. Duplication refers to the region that encodes the tyrosine kinase domain (TKD). These alterations are relevant with either low-grade extracerebellar astrocytomas of the cortex or dysembryoplastic neuroepithelial tumors [10]. Focal alterations of *FGFR1* are the most frequent alteration of the *FGFR1* gene in pediatric gliomas, affecting almost 30% of the patients [11] (Table 1).

##### PI3K-Akt-mTOR Pathway

The phosphoinositide 3-kinase (PI3K) cascade is implicated in crucial cellular stages like proliferation, apoptosis, motion, novel vascularization, and stem cell revitalization. The PI3K pathway is activated by growth factor receptors, such as epidermal growth factor (EGFR), EGFRvIII, platelet-derived growth factor (PDGFR), and RAS. It induces the production of phosphatidylinositol-3,4,5 trisphosphate (PIP3). Protein kinase B and Akt become operative by PIP3, activating a cascade that contributes to cell growth and debilitated cell death [12].

Phosphatase and tensin homolog (*PTEN*) dephosphorylates PIP3 and negatively modulates PI3K. Following Akt’s activation, the mammalian target of rapamycin (mTOR) exerts its metabolic impact on cellular growth. Aberrant and uncontrolled function of the PI3K pathway, due to mutations, occurs in neoplastic cells. Deletions of *PTEN* are observed in approximately 40% of gliomas [13] (Figure 3).

##### MAPK Pathway

Mitogen-activated protein kinases (MAPKs) are serine/threonine kinases that phosphorylate their own dual serine and threonine remainder to activate or deactivate their target [14]. MAPKs modulate cellular functions like metabolism, cell death, and immune responses [15]. The activation of the MAPK pathway takes place in the context of sequential phosphorylation. This setting includes a MAP3K (MAP kinase kinase kinase), which activates a MAP2K (MAP kinase kinase), which consequently activates a MAPK (MAP kinase) [14]. MAPK phosphorylation events can be inactivated by MAPK protein phosphatases (MKPs) [14]. There are three well-known MAPK pathways in mammalian cells. One of them is the ERK1/2, which responds to growth factors, hormones, and proinflammatory stimuli [16]. The function of both ERK1 and ERK2 variants commences after the attachment of a binding molecule to an RTK at the cell membrane. Accordingly, Ras protein, a member of G proteins, is activated. Ras incites the serine/threonine protein kinase, Raf, a MAP3K, which establishes the function of MAP2K or MEK, which finally adds phosphate molecules at threonine and tyrosine remnants of MAPK or ERK1/2 [16] (Figure 4). B-raf kinase is encoded by the *BRAF* gene located in 7q34. Mutations of this gene are frequently encountered in less malignant glial tumors and rarely in more malignant ones.

The Ras and PI3K cascades can be interchangeably modulated by various interrelated processes in different cellular phases. Therefore, Ras reacts immediately to the p110 fermentation segment of PI3K and induces the initiation of the PI3K cascade [17]. Furthermore, lengthened Ras activation decreases the encoding of PTEN, a phosphatase that dampens the PI3K activation [18]. Conversely, the PI3K-Akt cascade impedes the Ras cascade via the addition of a phosphate group on Raf1 [19].

##### MYB/MYBL1 Alterations

*MYB/MYBL1* mutations are regarded as discrete in the relevant studies. MYB protein production is detected in the nervous system [20]. MYB, along with the ubiquitous variant of the group MYBL2, modulates cyclin-dependent kinases (CDKs) production and function [20]. The *MYB* gene self-modulates its expression [20]. Excessive production of wild-type MYB protein is a promoter of tumorigenesis, and additional genetic alterations are needed for progression [21,22]. MYB adjuncts to the RNA-ligand protein QKI. It functions as a repositioned super enhancer of an active transcription factor [20,23]. *MYBL1* resettling is characteristic of low-malignancy glial cell tumors. 8q13.1 (which includes *MYBL1*) additions have been detected remarkably in diffuse astrocytoma grade 2 [24]. Lastly, transcription factor c-Myb binds and ignites the *MKP-3* gene transcription and downregulates the Ras-Raf-MEK-ERK cascade [25]. Gliomas harboring *MYB*/*MYBL1 mutations* are overall infrequent compared to other astrocytic tumors and recently included in the WHO grading [6]. Despite the variability in histopathology and clinical manifestations, they bear a good prognosis after adequate excision. Of note, all types of *MYB* or *MYBL1* mutations relate to the presence of supratentorial lesions resembling angiocentric gliomas (AG) [22,26].

#### 3.1.2. Types of LGGs

Pediatric-type low-grade gliomas (pLGGs) represent almost 10–20% of all cerebral neoplasias [26]. These malignancies are considered WHO grade 1 or 2 and develop anywhere along the neural axis. Common signs and symptoms include headaches, cognitive impairment, focal neurologic deficits, and personality or behavioral changes [26].

##### Diffuse Astrocytoma and MYB- or MYBL1-Altered

Diffuse astrocytoma with alterations of MYB/MYBL1 constitutes a separate category of tumors with indolent course and a very good prognosis [27,28]. This neoplasia is seen in the cerebral hemispheres. Patients are primarily young children, approximately 5 years old. Epilepsy, pyramidal deficits, and behavioral alterations are the main signs and symptoms [29]. Ten-year overall survival (OS) is 95.2%. The non-surgical treatment is recommended [28].

##### Angiocentric Glioma (AG)

AG is an infrequent cortical neoplasia (WHO grade 1) that occurs in adolescents and is usually manifested with seizures. Its molecular identity is *MYB-QKI* gene fusion [30]. This mutation responds to almost 90% of AGs and in less than half of pediatric DLGGs. The result of this alteration is the deficient onco-suppression of *QKI* accompanied by the intense expression of MYB. The tumor demonstrates indistinct boundaries [31].

##### Polymorphous Low-Grade Neuroepithelial Tumor in Youth (PLNTY)

PLNTY is a seizure-relevant neoplasia with alterations in the MAPK pathway [32]. Molecular profiling reveals a *BRAF V600E* focal alteration (≈40%) or an *FGFR 2/3* fusion (≈50%) [33].

##### Diffuse Low-Grade Glioma—MAPK Pathway-Altered

This is a recently described lesion that harbors both *BRAF V600E* mutation and FGFR1. This glioma is classified into the intermediate risk group [34].

##### Pilocytic Astrocytoma (PA) (Figure 5)

Pilocytic astrocytoma is the most frequent cerebral neoplasia of childhood and adolescence, and its incidence rate is approximately 15%. It usually develops in the first two decades of life [35]. The term pilocytic originates from the histologic features, which include thin, elongated astrocytic projections stained with GFAP or cells with hair-like, bipolar processes [36]. The cerebellum is the most frequent site of development (43%), followed by the convexity areas (36%), the visual tract and hypothalamus (9%), and the brainstem and spinal cord [36]. Three histologically distinct patterns exist: (1) The two-compartment type is very frequently detected in cerebellar PA; it involves cellular areas with saccules, glassy vascular networks, and scarce aggregations of endoplasmic reticulum with rosettes. (2) The compact pattern is mainly composed of bipolar cells with GFAP-positive fibers. (3) The not firmly arranged pattern contains cells carrying one axon and many dendrites. Microscopic infiltration of the leptomeninges is frequent. In the case of the optic nerve PA, the invasion occurs in the subarachnoid space, and the optic nerve is significantly enlarged. In the genetic field, PA bears typical molecular alterations of the MAPK pathway [36].

**Figure 5 cancers-17-01566-f005:**
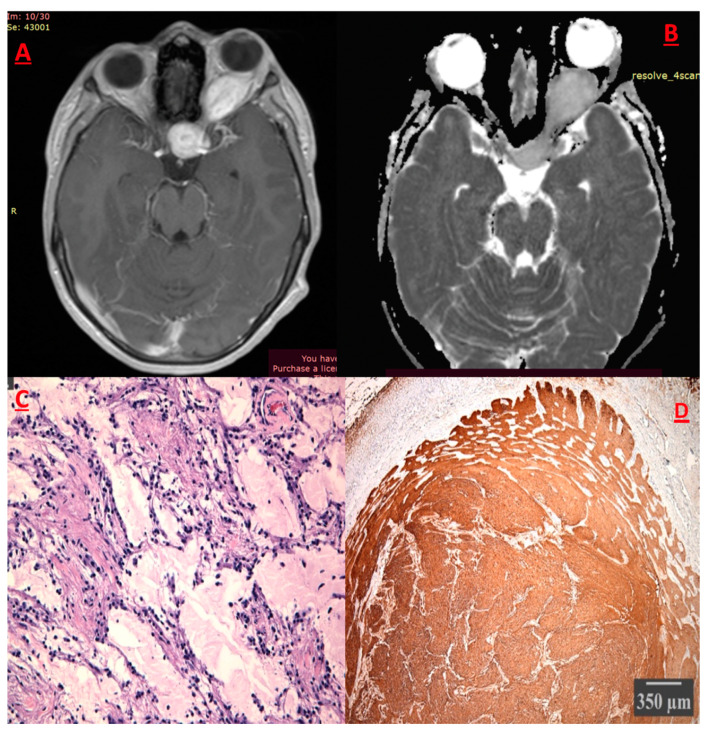
Optic pathway glioma. (**A**) T1 Sequence with contrast medium enhancement of Dumbbell-like glioma of left optic nerve. (**B**) tumor exhibits moderate diffusion of water molecules in the ADC sequence. (**C**) Hematoxylin–eosine: biphasic pattern with eosinoplic granular bodies, lack of Rosenthal fibers and myxoid cystic background. (**D**)Epithelial membrane stain: Optic nerve infiltration, remarkable subarachnoid space thickening and meningothelial excessive proliferation.

PA strongly correlates with Neurofibromatosis type 1 disease in such a way that one-fifth of NF1 patients develop PA regarding the visual tract [37]. The most frequently encountered mutation, though, is the *KIAA1549-BRAF gene* fusion [38]. This mutation is an internal tandem copy in the 7q34 locus. The incidence of the mutation varies overall from 50–70% and responds to cerebellar lesions (90%) [38,39]. In addition, a common somatic alteration of the *BRAF* gene is the V600E substitution, which is strongly associated with extracerebellar sites of development. *FGFR1* mutations are majorly confined to midline areas. Finally, combined *BRAF V600E* and *NTRK* group gene hybridization are frequent in supratentorial neoplasms [36].

The *NF1* gene on 17q11.2 expresses neurofibromin, an onco-suppressor protein, which also downregulates the MAPK cascade (Figure 4). Fifty percent (50%) of the patients with NF1 harbor a pathogenic germline mutation from a parent, while the other half represent a sporadic novel mutation [40]. NF1-associated tumors occur when the NF1 gene is bi-allelically inactivated [41]. This inactivation results in the excessive function of the RAS/MAPK signaling cascade. BRAF mutations are rare in PAs, but when they are present, they relate to cerebellar location [41]. Clinical manifestations depend on the location of origin [35]. In the field of imaging, two broad patterns exist: (1) expansive solid lesions with scant saccular regions of variable size. The lesion usually exhibits negligible surrounding edema. The lesions usually exhibit negligible surrounding edema and may emerge in the cerebellum or the hemispheres. (2) Infiltrative PA is usually located in the optic-diencephalic region or brainstem [42] (Figure 5). PAs, therefore, constitute a neoplasia with common clinical and histological characteristics, rather than a distinct tumor type. Molecular profiling aims to prompt diagnosis of these tumors. Despite their assumed variability, their treatment and strongest prognosticator is surgical excision.

The pilomyxoid variant (PMA) is contained in the body of the pilocytic astrocytoma and is considered a WHO grade II malignancy. PMAs arise mostly in the hypothalamic/chiasmatic region in very young children. Some pilomyxoid neoplasias relapse as classical PA, which may constitute the primary phase of PA evolution [36]. PMA is also called the “infantile form” of PA because it is often recognized at the early stages of life. Radiologically it has similar features to PA, but without the cystic part, and it has a higher incidence of leptomeningeal dissemination. In the molecular field, PMA is characterized by BRAF gene fusion and chromosomal losses in contrast to PA [43]. Treatment of PA is surgical resection, targeting the solid and contrast-enhanced part of it [35,42]. There is a controversy about removing the cystic wall. It has been suggested that when the cyst wall is enhanced in MRI imaging, the surgeon should proceed to its removal if possible. When tumor recurrence is encountered, the preferred method of management is re-operation. However, when this is not a feasible option, then chemotherapy is applied. Radiotherapy is also an option in recurrent episodes when surgery is not a plausible option. This is the least alternative, due to its consequences (endocrinological derangement, vasculopathy, cognitive decline, and the occurrence of post-irradiation malignancy) [35]. In the case of brainstem gliomas, stereotactic biopsy is the preferred method for histologic identification, while radiotherapy is the first line of treatment [35]. When complete excision is not feasible, there are several reports that PA remnants may stop growing. This phenomenon is probably attributed to vascular interruption and cellular apoptosis during surgical manipulations. Therefore, in those cases a “watch and wait” policy is recommended with a surveillance imaging plan at 12, 18, 30, 42, and 66 postoperative months [42].

##### Pleiomorphic Xanthoastrocytoma (PXA)

Almost half of individuals with either PXAs (WHO grade 2) or anaplastic PXAs (WHO grade 3) carry the *BRAF V600E* mutation (leucine substitutes valine at the aforementioned position within the gene) [44]. It has been suggested that PXA derives from astrocytes under leptomeninges, justifying their cortical emergence [45]. PXA is very often encountered in adolescence. Temporal location is the most frequent scenario, accounting for <1% of all astrocytomas [46]. *BRAF V600E* is seen in other brain malignancies combined with other mutations, though [47]. *BRAF V600E* seems to be the most frequent genetic alteration [48]. The overall 10-year survival is 67%. PXAs occur mostly in teenagers and adolescents. The frequency of their emergence is sequential in the temporal, parietal, and occipital lobes. Seizures are the main clinical sign [49]. Despite the fact that PXA is a non-aggressive lesion, it tends usually to relapse and to deteriorate into a high-grade tumor. Relapse rate is approximately 30% and progression rate 10–20% [45]. Infratentorial location of the lesions seems to have higher rates of recurrence, approximately 47% [50]. Histologically, PXA seems to bear features of moderate cellularity without any necrosis and rare mitosis [51]. A histologic hallmark of PXA is the superficial location of the lesion with leptomeningeal extension. It engulfs cortical vessels rather frequently [45]. On CT and MRI, the scalloping of the inner table of the skull bone is a frequent finding [52]. Clinical manifestations of PXA most frequently include long-standing headaches and focal neurological deficits, with seizures occurring in 58.8% of the cases. They are attributed to the temporal location [53]. The gold standard of care for PXA is excision and post-procedural chemotherapy in cases of remaining disease or relapse. Complete resection seems to achieve a decade overall survival rate of 82% of cases [45].

##### Dysembryoblastic Neuroepithelial Tumor (DNET)

*FGFR1* point alterations or duplications are isolated in most DNETs. These are classified as dysplastic low-grade tumors. The clinical stability of DNET is being reappraised. True neoplastic development appears in cases of residual lesions. In cases of refractory epilepsy, a re-operation is warranted. Dysplasias are usually situated in the temporal and frontal regions. They appear in older children and adolescents [54].

##### Subependymal Giant Cell Astrocytoma (SEGA)

SEGA harbors either *TSC1* or *TSC2* homozygous inactivation in about 80% of cases. Therefore, the tuberin–hamartin protein complex is deactivated, whereas the PI3K cascade is hyperactivated [55]. Miscellaneous genetic aberrations, such as partial loss of Chr22 and *BRAF V600E* mutations, have been reported [56]. These tumors evolve within the first twenty years of life. They exhibit rather slow progression and arise intraventricularly. Approximately 10% of the patients suffer from tuberous sclerosis [56].

### 3.2. High-Grade Gliomas (HGGs)

HGGs are grade III and IV gliomas. They comprise 15–20% of all pediatric cerebral neoplasms. Their 24-month survival rate is confined between 10 and 30%. These tumors do not undergo malignant evolution from LGG, in contrast to adult HGGs. They carry mutations in the genes expressing histone H3.3 and H3.1 [57] (Figure 6). Histones are protein complexes that are a part of the nucleosomes and regulate chromatin wrapping. Oncogenic mutations in histones, now termed “oncohistones”, are the molecular hallmarks of pediatric HGGs and appear in histone H3 [58]. There is a characteristic spatial association between specific mutations and the anatomic site of development. The outcome of these mutations is the derangement of the methylation process, which usually results in global hypomethylation states [59].

#### 3.2.1. Molecular Substrate

##### Histones and Nucleosomes

DNA of higher eukaryotes is found tightly swathed around an eight-histone complex (each of H2A, H2B, H3, and H4 are duplicated) comprising a nucleosome (Figure 7). Actually, half of the mass of a chromosome is formed by these proteins. This configuration does not only provide packaging, but also it participates in the epigenetic regulation of gene expression as well [60]. Two phases of chromatin interchange: the unwrapped euchromatin, which is liable to transcription, and heterochromatin. The latter corresponds to a dense DNA–histones complex not amenable to transcription [61].

Three major categories of H3 exist: (1) duplicated H3.1 and H3.2 conjuncts, which are named “canonical histones”, whose zenith production occurs throughout the S-phase. (2) The replacement variant H3.3. Its expression is irrelevant to the cell cycle phase and regulates the action status and maintenance of chromatin. The DAXX-ATRX complex integrates H3.3 into heterochromatic regions. Furthermore, H3.3 and ATRX participate in telomere perseverance throughout cell lineages differentiation [58]. H3.3 deposition relates to enhancers, promoters, and gene loci as well. Lastly, (3) the third category is the centromere protein [62].

##### DNA Packaging and Methylation Control

DNA methylation sustains the distinct genes’ expression throughout the cell cycle [63]. It actually confers a chromatin “silent” condition together with the proteins that configure nucleosomes [64,65]. Proteins responsible for DNA methylation act on specific CpG sites of the genome.

At this level, cancer initiation interrelates with mutations impairing the action of enhancer of zeste homolog 2 (EZH2), which participates in H3K27 methylation and is a part of polycomb repressive complex 2 (PRC2). Therefore, *EZH2* fusions lead to hypermethylation of H3K27 [66]. On the other hand, deletions of *EZH2* relate to hypomethylation of H3K27 target genes [67]. On the contrary, the switch/sucrose non-fermentable (SWI/SNF) protein group, otherwise encountered as the BRG1/BRM-related factor (BAF) complex, modulates transcription by reshaping chromatin using ATP [68]. Two other variant complexes also exist: the polybromo-associated BAF (PBAF) and the non-canonical BAF (ncBAF). This group of proteins facilitates DNA restoration, transcription, and nucleosome arrangement [69] (Figure 8). The transcription profile of genes that contribute to human hindbrain formation (e.g., *HOX* genes) is epigenetically regulated by the trimethylation of H3K27M. Deficient trimethylation of H3K27M, either due to histone alterations or EZH2 mutations, leads to a loss of the aforementioned transcription factor silencing. This molecular procedure occurs, especially in progenitor cells, and, thus, the tumorigenesis may be explained in areas like the brainstem or posterior fossa [64,66,69].

IDH1/2 mutations have an impact on a Krebs cycle. These mutations suppress the demethylating activity of these enzymes [70]. These alterations are rather infrequent (~6%) in children’s high-grade gliomas, and they emerge in the telencephalon at late adolescence. IDH and ATRX alterations coexist frequently in the same way there is an overlap between IDH and H3.3 alterations [70] (Figure 9A,B).

Telomerase is a transcriptase that identifies and elongates the telomeres (TTAGGG sequences) [71]. The TERT gene produces the functional enzymatic group of telomerases [71]. Telomerase activity ceases in non-neoplastic cell groups, and telomeres decrease in length throughout the consecutive cell cycles. In the end, upon a crucial telomere extent, cells start senescing. The vast majority of neoplasms encode this transcriptase [72] (Figure 10).

Alpha-thalassemia X-linked intellectual disability (ATRX) is a chromatin remodeler. ATRX acts in the setting of the DAXX/ATRX complex, which deposits the H3.3 histone isoform into the compact chromatin areas. Its inactivation leads to the alternative lengthening of the telomere (ALT) that is routinely performed by the telomerase reverse transcriptase (TERT) expression. Therefore, ATRX inactivation allows the perpetual division of cancer cells and, thus, the accumulation of additional mutations [12]. It belongs to the SWI/SNF group [73]. The deposition of H3.3 facilitates the trimethylating at the K9 position (H3K9me3) and inhibits transcription [74]. This deposition is situated at the telomere point and participates in DNA damage repair. DAXX acts as an ATP-unrelated satellite that deters pathological protein accumulation [75] (Figure 11A,B).

SETD2 restores chromatin and maintains transcription by methylating histone H3 on lysine 36 (H3K36). Mutations on H3G34 (arginine or valine instead of glycine) modify the transcriptional and epigenetic status [76] (Figure 9B).

The enzyme O-6-methylguanine-DNA methyltransferase (MGMT) inhibits the addition of alkyl groups at the O6 location of guanine [77]. The encoding of MGMT is inversely related to the (CpG) islands’ hypermethylation along the MGMT gene promoter. This fact confers liability to the chemotherapeutics, such as temozolomide (TMZ), which methylates this location and deters the DNA replication [78].

##### Cell Phases Control

The cell’s sequential phases secure the equal allocation of the genome into two descendant cells. The consecutive phases are actually four: G1 phase (gap1 phase), S phase (synthesis phase), G2 phase (gap2 phase), and M phase (mitosis) [79]. After cycle completion, cells may transit to an inert phase recognized as G0. At the G1 phase, D-type cyclins (D1, D2, and D3) are encoded and activate the Cyclin D-CDK4/6 complex. The latter deactivates, through phosphorylation, the retinoblastoma protein (pRb), which detaches from E2F transcription factor. This factor, together with Rb, previously constituted a complex that hindered the passage from the G1 phase to the S phase [79] (Figure 12).

##### Cyclins and Cyclin-Dependent Kinases (CDKs)

Cyclins are supervisory protein substrates with high specificity for their fermentation partner CDKs. The latter are proline-directed serine/threonine-protein kinases. They have a two-lobed configuration and modulate the shift during the cell phases [82]. The alterations undergo the supervision of transcription factors (TFs) or regulatory proteins like pRb, which suppress transcription [83]. The retinoblastoma 1 (*RB1*) gene is located at 13q14.p16. It is also encountered as either *p16INK4* or *CDKN2A*. Loss of function of pRb leads to the disinhibition of proteins necessary for either the S period or mitosis [83] (Figure 13).

##### Check Point Proteins

The cyclin-dependent kinase inhibitors constitute a group of cell phase supervisors as well. The major activity is the configuration of secure complexes, along with the impedence of the cell phase transition. Prominent proteins, such as p15, p16, p18, and p19, attach firmly to CDK4/6 and hinder the CDK4/6–Cyclin D interaction [84].

p15 and p16 hinder the addition of phosphate molecules to Rb and passage to the S-period [80]. p16 is an onco-suppressor protein expressed by the *INK4a/ARF* locus of chromosome 9p21 [85]. p16 modifies Rb, and their expression is inversely correlated [86]. Rb1 modifies E2F transcription factors. When Rb1 is not impaired, D-type cyclins combined with CDK4/6 add phosphates to Rb1, and the latter detaches the inhibitory E2Fs [87] (Figure 12).

The *p53* gene lies at 17p13.1 and encodes TP53 protein. All normal cells express this, with its normal half-life protein being minimal. The TP53 protein‘s function concerns the sequence-based transcription. Alterations of the *p53* gene are usual in high-grade malignancies. The *p53* mutations respond to patients with more brief disease-free phases and poorer outcomes [86,88].

Almost 40% of HGGs in children relate to alterations in the *p53*, and in contrast to high-grade gliomas in adults, those HGGs in childhood have been correlated with improved 5-year progression-free survival (PFS) [89].

#### 3.2.2. Diffuse Pediatric Midline Glioma (DMG)

This tumor is a WHO grade 4 astrocytic lesion of the brainstem or the midline. These pediatric tumors are typically located in the pons (diffuse intrinsic pontine glioma—DIPG), or they may extend to both thalami [90]. In adolescents and adults, they invade only one thalamus or the spinal cord, respectively [91]. DIPG bears the worst prognosis, with a 2-year survival of less than 20% [92]. The hallmark is the loss of trimethylation at H3K27 and is subclassified into three types: (1) DMG, with H3K27 alterations of either H3.3, H3.1, or H3.2-variant; (2) DMG, H3-wild-type without K27 alterations, but excessive function of enhancer of zeste homologue inhibitory protein (EZHIP) and hypomethylation at K27 location; and (3) DMG, with EGFR mutations [93]. The DMG H3.3 K27-mutant exists in children 7–8 years old and lies in the midline or the pons; its median OS counts 10 months. A total of 25–50% of DIPG carry K27M aberration of the H3.1 isoform [59].

H3.1 and H3.2 K27–altered DMG appear exclusively in the pons of children aged 5 years old, and its OS rate is approximately 15 months [94]. The DMG H3K27-mutant, compared to the DMG H3-wild-type, characteristically exhibits both hypomethylation and hyperacetylation at the H3K27 location [95]. H3 K27M mutations restrict PRC2 activity by sequestration of its catalytic subunit (EZH2) [96]. PRC2 target genes are involved in developmental processes, and, as a result of hypomethylation, they are all upregulated, and its tumor-suppression action is diminished [95]. Generalized deficiency of H3K27me3 overlaps with a reciprocal overall excess of H3K27 acetylation that is copious in the proximity of bromodomain-containing proteins BRD 2/4. The latter are supervisors of RNA Polymerase II throughout the transcription. Therefore, H3 K27M acetylation presumably enhances transcription [58,97]. The loss of H3K27 trimethylation coexists with BRAF V600E substitution and, less commonly, with IDH1 alterations [93]. The H3-wild-type DMG is associated with EZHIP excessive production. It is not so frequently encountered and emerges in common cerebral areas and within the same time spectrum as the H3.3-altered type [93,98]. EGFR-mutant DMG invades the thalami bilaterally. Its target group is school-aged children [99].

#### 3.2.3. Diffuse Pediatric Hemispheric Glioma

Prognosis is generally ominous. Median survival ranges from 9 to 15 months, and hardly one-fifth of the patients survive up to 60 months [12]. Other than ionizing radiation, no other environmental risk factor has been identified so far [100]. Genetic inheritance is implicated in pediatric high-grade gliomas, such as NF1, Turcot syndrome, and Li–Fraumeni syndrome [92].

This tumor carries the H3 G34 mutation. There are two subtypes, according to the amino acid interchange during H3 expression: the G34R (arginine substitutes glycine) or the less frequent G34V (valine substitutes glycine) [101]. It is a highly infiltrative WHO grade 4 lesion that originates in the forebrain structures, especially in the temporal and parietal lobes. The median age of occurrence is the middle of adolescence [94].

TP53 coexists in about 40% of cases, while ATRX and (DAXX) alterations usually co-occur; the MGMT promoter (O6-methylguanine-DNA methyltransferase) is frequently methylated [100,101]. Furthermore, 80% of these tumors disclose excessive function of the PI3-Kinase/Akt/mTOR cascade. This is attributed to PTEN promoter methylation [92]. Approximately 60% of this type carry *PDGFRA* alterations [102]. Finally, H3.3 G34R/V has been somehow related to MYC/MYCN robustness, and when that happens, MYC upregulation occurs through differential H3K36me3 binding. The MYCN upregulation is considered a driver of tumor development [94]. In hemispheric pediatric HGGs, K36 trimethylation (K36me3) is decreased. In general, K36 can only be methylated with three methylate groups by SETD2 (SET domain containing 2 protein) methyltransferase. SETD2’s normal function is the junction of histones’ flexible appendix to a protein’s furrow. This process fails in the case of H3.3 G34R/V mutation due to the fact that protein’s furrow does not engulf an amino acid greater than glycine [12]. The glioblastoma-like type exhibits massive expression of glial fibrillary acidic protein (GFAP) [102]. The median survival is hardly two years. The MGMT promoter methylation portends a better outcome, whereas amplification of oncogenes (e.g., EGFR, CDK4, and MDM2) relates to an ominous prognosis [101]. The isolations of PDGFRA mutations in half of patients confer innovative treatment chances [102]. Molecular profile correlates strongly with location [103] (Figure 14).

#### 3.2.4. Diffuse Wild Type High-Grade Gliomas

These gliomas exhibit methylation variations and are classified as pHGG (receptor tyrosine kinase) RTK1 (with copious *PDGFRa* amplification) or pHGG RTK2 (with copious *EGFR* amplification). Finally, a subgroup with *n-Myc* amplification has been observed [104].

The great proportion of these tumors arise supratentorially. A small part of pHGG MYCN (responding to 15%) appears in the brainstem [101]. pHGG RTK1 emerges both infratentorially and in structures within the brainstem [101]. The outcome has been considered dismal, and median OS counts for almost one and a half years. On the contrary, RTK2 pHGG interrelates with a median OS of 44 months and pHGG MYCN 14 months [105]. Pontine tumors are the most malignant of HGG-MYCN, and OS rates are 1.5 months [106]. *PDGFRA* amplification commonly results in activation of the PI3K/mTOR or MAPK cascades. They relate to worse prognosis [107].

pHGG RTK1 type encompasses the majority of irradiation-associated gliomas [106]. The occurrences of TP53 and ATRX mutations are associated with adverse outcomes [108]. pHGG BRAFV600E mutants (WHO-grade 3 PXAs) have a 24-month OS in 67% of the cases [94]. They coexist, however, with alterations like CDKN2A/B deletion or TERT promoter mutations. The aforementioned alterations result in benign tumors with ominous outcomes [109].

#### 3.2.5. Infant-Type Hemispheric Gliomas

These gliomas include three types: (1) hemispheric (RTK)-promoted malignancies, including anaplastic lymphoma kinase (ALK), c-ros oncogene (ROS1), neurotrophic receptor kinase 1 NTRK, and mesenchymal epithelial transition (MET) fusions. This group exhibits a moderately poor outcome, (2) hemispheric *RAS/MAPK*-related lesions that demonstrate prolonged OS under short-term adjuvant therapy, and (3) midline *RAS/MAPK*-related neoplasms with a rather dismal prognosis in the setting of novel chemotherapeutic protocols [110].

ALK mutations align with LGGs (10-year OS 53.8%), whereas ROS1/MET (25%) and NTRK fusions (42.9%) align with HGG. The second group does not include BRAF mutations and has the best 10-year OS among the three groups (93%). The dismal prognosis of BRAF-fused midline neoplasms up to the first year of life is remarkable, and their behavior is completely different than in older age groups (5-year OS rates 23%). This discrepancy is imputed to both age-related genetic and microenvironment agents [110]. *BRAF V600E* alterations in PXAs have been extensively reported. PXAs tend to upgrade to more malignant lesions, especially when they combine with *CDKN2A* deletions [111]. Interestingly, the frequency of BRAF alterations in pHGG varies from 10% to 25% [112]. Therefore, BRAF alterations in the high-grade lesions probably exist in the setting of secondary pHGG [113].

### 3.3. Current Treatment Approaches

Pathology of pLGG constitutes an independent prognosticator of outcome. Therefore, the well-demarcated PAs are amenable to complete excision [114]. Extent of surgical excision is the predictor for prolonged progression-free and OS outmatching pathology [115,116]. The 60-month progression-free survival (PFS) varies from 75% to 100% of cases [117]. Regular clinical and radiological reevaluations are necessary. Chemotherapy is administered in case of recurrence [118]. Postsurgical irradiation following subtotal excision provides longer PFS, but not OS [119]. The precise involvement of BRAFs in PA is not clear; however, its inhibitors participate in the therapeutic planning [120].

High-grade glial-originating neoplasms constitute a hugely inhomogeneous category with specific alterations and prognoses, which warrant tailored targeted therapy. Surgery for pediatric HGGs has three primary objectives: (i) obtain samples for identification, (ii) relieve elevated intracranial pressure, and (iii) tumor cytoreduction. Stereotactic needle biopsy is indicated for deep or ineloquent areas seated malignancies. Surgery is followed by radiotherapy and temozolomide administration. This schema does not markedly contribute in the pediatric population, though [121]. The 60-month OS is confined to 20%, and the median OS is between 12 and 42 months [122].

Children with GBMs and MGMT promoter methylation exhibit 13.7 months OS in contrast to hardly three months for those that do not harbor the methylation [123]. Excessive activation of Akt shortens 12-month failure-free survival compared with gliomas with none [124].

Both location and consequent prognosis correlate with molecular profile. *PDGFRa* usually undergoes amplification [125]. Non-mutated *PDGFRa* lesions occur in the brainstem, whereas *PDGFRa* somatic alterations predilect tumor emergence far from the brainstem [126]. Frequently, pHGGs are erroneously characterized as pLGGs. This had been the rule with pLGGs according to the previous WHO classifications, which did not evaluate H3.3K27M mutations. Another reason is the limited and insufficient tissue sampling, which excludes malignant areas, in cases of non-homogeneous lesions. Overall PLGGs carrying either H3.3 K27M or TP53 alterations constitute high-grade-behaving gliomas [113].

In case of recurrent pediatric HGGs, besides the fact that prognosis remains unfavorable, a palliative treatment plan has to be established. Another schema of radiotherapy with 30–54 Gy confers clinical stabilization and survival enhancement even in DIPG cases [127].

## 4. Embryonal Tumors

This group involves tumors whose microscopic characteristic is the presence of small round cell populations. This tumor group is, in fact, genetically diverse [128]. Medulloblastomas, atypical teratoid/rhabdoids, and embryonal tumors with multilayered rosettes are the main representatives (Figure 2).

### 4.1. Molecular History

#### 4.1.1. E-Cadherin/Beta-Catenin Complex

The wingless (Wnt) signaling hinders the addition of phosphates and decomposition of the beta-catenin protein. The latter aggregates in the cytoplasm and translocates to the nucleus. It acts as a transcription cofactor and enhances cell proliferation [60]. Decreased activation of the E-cadherin/beta-catenin complex implies high malignancy [60].

Alterations of the *CTNNB1* lead to the accumulation of b-catenin and bear resemblance to the excessively functioning Wnt pathway [60] (Figure 15).

#### 4.1.2. Sonic Hedgehog Pathway (SHH)

The sonic hedgehog (SHH) canonical pathway initiates when the glycoprotein SHH attaches to and impairs the functioning of the transmembrane protein, Patched 1 (PTCH1). When the SHH-binding molecule is absent, the transmembrane protein smoothened (SMO) is inhibited by PTCH1. Hence, the joined molecule of SHH/PTCH1 modulates SMO function [129]. SMO is a GPCR-like (G protein-coupled receptor) protein, and its translocation into the cilia membrane activates Gli [129]. In response to SHH signaling, PTCH1 inhibition of SMO is abolished [130]. When PTCH1 is decomposed, SMO aggregates and incites the downregulation of the SHH pathway (Figure 16).

This pathway enables Gli proteins to enter the nucleus. Expression of the *Ptch1* gene enhances transcription, whereas encoding of *Gli1* hinders it [131]. Vice versa, GLI entrance to the nucleus regulates the expression of WNT and Noggin [131]. The SHH cascade is also regulated by Suppressor of Fused Protein (SUFU). When the SHH ligand is absent, SUFU attaches immediately to GLI proteins and impedes their presence at the nucleus [131].

#### 4.1.3. Medulloblastomas (MB) (Figure 17)

MBs represent a diverse category of neoplasms. Normal development and tumorigenesis share changes in cell growth, migration, and death [132]. MBs are assumed to emanate from precursor cells of the external granular layer (EGL) of the hindbrain [133]. They are initially located around the rhombic crest during organogenesis. Granular neural cell growth is driven by the SHH pathway [134]. One out of two MBs reveals the aberrant initiation of the SHH or Wnt cascades [135]. A good interaction between them regulates the neural crest formation [136]. These pathways induce expression of MYCN. Excessive MYCN encoding incites development and transposition of neuroblasts, while an attenuated encoding leads to terminal differentiation [137]. According to a 2012 international consensus study on molecular analysis of medulloblastomas, four distinct types with different prognoses were introduced. These involved MBs with aberrancies of WNT cascade, SHH pathway, Group 3, and Group 4 [138] (Table 2). More recently, WHO advocates the usage of WNT, SHH with and without TP53 mutation, and non-WNT/SHH molecular groups [139].

**Figure 17 cancers-17-01566-f017:**
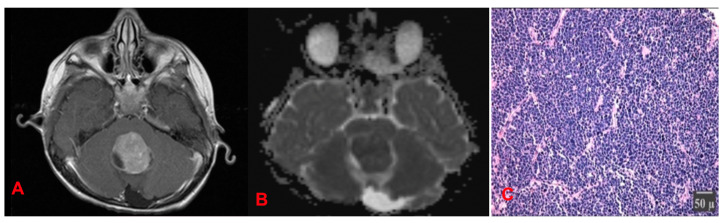
(**A**) MRI. T1 sequence. Midline vermian intraventricular tumor, which enhances after contrast medium administration. (**B**) Increased diffusion in the DWI sequence indicative of high cellularity. (**C**) Hematoxylin–eosin stain: Areas of high cellular concentration. Cells are majorly spherical, with an increased nuclear–cytoplasmic ratio.

WNT-activated MBs are the most infrequent (10%) and are observed in late adolescence. They are seldom encountered in ages below one year [140]. These tumors emanate from the inferior rhombic crest and dorsal rhombencephalon. They tend to occupy the cerebellopontine angle and lie along the lateral foramen, cisterna magna, or fourth ventricle [141]. Molecular detection of *CTNNB1* mutation is recommended. The great proportion of these lesions coexist with monosomy 6. Additional ignition of the SHH cascade is sometimes documented [142]. It can coexist in the context of Turcot syndrome (10%) when APC mutations exist [143]. They reveal excellent prognosis in children, and metastatic disease is rare. They hardly recur, and their 5-year OS is greater than 90% [142].

SHH-activated MBs (SHH MBs) comprise 30% of all MBs. They emanate from cerebellar granule neuron precursors and are located in the cerebellum [142]. Apart from the SHH pathway activation, this group may carry the TP53 mutation, which has repercussions for prognoses. MBSHH has a great variance of forms [144]. SHH MB wild-type TP53 has a two-peak age distribution. It concerns children younger than one year of life and the period shortly after adolescence. No differences in sex distribution exist [145]. Extensive nodularity in histology is characteristic and concerns infants. PTCH1 or SUFU alterations are usually responsible for the SHH cascade ignition [143]. Patients with Gorlin syndrome, who carry SUFU mutations, are liable to MB occurrence at a young age [145]. SHH MB with altered TP53 is infrequent and manifests in school-age children and teens. They have a worse prognosis with a poorer response to therapy. On the contrary, TP53 wild-type tumors during infancy are low-risk [142]. Germline TP53 mutation detection is included in the diagnostic workup when Li–Fraumeni syndrome is suspected [145].

Group 3 MB concerns almost one-fifth of MBs. Infants constitute one-half of the affected population, with a preponderance for the male sex. In almost half of the cases, metastases exist [138]. This group is hardly observed in adults [145]. These tumors originate from neural crest cells and are ordinarily situated in the cerebellar vermis near the fourth ventricle [146].

A variant of MB3 reveals increased copies of *c-Myc*, which signifies an ominous outcome [142] (Figure 18). Additional molecular alterations involve SMARC4, a subunit of the SWI/SNNF complex [142,143]. This group is linked with the worst prognosis because of the tendency to metastasize and *c-Myc* excessive expression [142].

Group 4 comprises 40% of all MBs. These tumors appear predominantly in school-aged children and young adolescents. Males have a threefold higher risk of developing these tumors [145]. They emanate from the superior rhombic crest and extend to the cerebellar vermis [97]. MB4 coexists with isodicentric chromosome 17 and manifests *n-Myc* amplification [143] (Figure 18). They tend to metastasize, and this is the main prognosticator [146].

#### 4.1.4. Treatment Status

##### Radiation Therapy

Adjunct to surgical excision RT led to advances in the survival of patients. The 5-year OS counts are 50–65% with RT alone [147]. No general agreement exists about the cut-off age regarding the delay of employment of craniospinal RT. Ranges vary from 3, and in some studies, to 4 or 5 years old [147]. Treatment with proton beams may confer reduced long-period toxicity. In the short-term, though, demarcated necrosis, especially in the brainstem, exists [148].

##### Chemotherapy in Standard-Risk Patients

The long-term sequelae of this treatment are remarkable, especially in preschoolers and school-aged children. Combined chemotherapy and radiotherapy result in significantly lower doses of craniospinal radiation [148]. The results revealed a favorable 60-month OS in 79% of the patients, compared to studies without chemotherapy up to that time. Furthermore, long-period neuropsychological assessments may improve intellectual outcomes [147].

### 4.2. Atypical Teratoid/Rhabdoid Tumors

This group is considered a highly aggressive neoplasm. Its hallmark mutation regards the chromatin modifier group B member 1 (SMARCB1) that constitutes a subunit of the SWI/SNF molecule (Figure 8). These tumors appear in infancy and early childhood. They exhibit very poor survival [149]. Recently mutations in SMARCA4 have also been described [150].

Four subtypes are described based on the methylation profile [151].

The TYR subtype relates to the monosomy of chromosome 22 and point alterations of the *Smarcb1* gene. This group is predominantly located at the posterior fossa and demonstrates hypermethylation of their DNA.

The SHH subtype, which is imputed to either focal deletions or point mutations of the *Smarcb1* gene, has the greatest incidence among all groups. These lesions are frequently situated supratentorially and present hypermethylation in their DNA as well. They carry abnormal SHH signaling.

The MYC subtype is characterized by broad deletions of the *Smarcb1* gene. The *Myc* genes are overexpressed. It has almost equivalent presence in both supra- and infra-tentorial areas.

The SMARCA 4 subtype is the least common. Its features are the point mutations of *Smarca 4.*

Reported therapeutic protocols involve combined treatment with resection, irradiation, and chemotherapy. The latter involves either the conventional protocols or high-dose chemotherapy (HDC). In this case autologous hematopoietic progenitor cell rescue (AHPCR) should precede, and intrathecal (IT) administration should be a part of the protocol. Significant presupposition remains, though, about the degree of the malignancy’s excision [150].

### 4.3. Embryonal Tumor with Multilayered Rosettes (ETMR)—C19MC-Altered) and Embryonal Tumor with Multilayered Rosettes—Not Otherwise Specified

#### 4.3.1. ETMR Tumors

ETMR is a group of tumors whose origin is attributed to the presence of chromosome 19 microRNA clusters (*C19 MC*). The group manifests either excessive copies of the respective gene or annexation with the *TTYH1* gene. This novel group engulfs the older entities of embryonal tumor with abundant neuropil and true rosettes (ETANTR), ependymoblastoma, and medulloepithelioma [152]. They occur in toddlers or even in younger groups [142]. They appear predominantly in cerebral hemispheres and occasionally extend to both of them. They may carry cystic regions and calcifications. Their histological hallmark is the true rosettes accompanied by neuroblasts [142]. ETANTR contains areas of embryonal minuscule stratified spherical blue cells and true rosettes, combined with scant cytoplasmic neuropil domains [153]. Ependymoblastomas also contain multiple stratified rosettes, primitive cells, and sometimes fibrillary processes. They lack neuropil. Medulloepitheliomas do not exhibit a significant amount of neuropil. Mature neurons and astrocytes intermingle with embryonal cells [142].

Except for *C19 MC* mutations, the gaining of chromosome 2 is usually encountered [142]. The lin-28 homolog A antibody is an immunological substitute for C19 MC alterations; however, it is not specific [154]. Further molecular profiling showed biallelic alterations of the *DICER1* gene (Figure 19). The outcome is poor. They relapse and often metastasize [155]. The gold standard of treatment involves the maximum safe resection and chemotherapy; albeit, the median OS remains one year [155]. Radiotherapy, in particular proton therapy, has recently been shown to prolong survival [156].

#### 4.3.2. Pineoblastomas (PBs)

These are excessively invasive pineal neoplasms with poorly delineated tumor margins. They appear within the first twenty years of life. They exhibit embryonal tumor morphology with true rosettes and sometimes variable differentiation [158]. Epigenetically, pineoblastoma is inhomogeneous and is composed of five separate fundamental molecular subgroups with distinct courses [159]. These subgroups demonstrate aberrant microRNA transcription. Succinctly, they are *the DICER1*, the *DROSHA (DGCR8)*, and the *KBTBD4* subgroups [160]. Older children are usually affected, and the outcome is good after irradiation (5-year OS is 60–100%) [161]. The remaining two subgroups are *PB-MYC/FOXR2* and *PB-RB1.* Their distinctive feature is the oncogenic MYC-miR-17/92-RB1 circuit. They concern infants and young children. Irradiation is, thus, deterred and associated with bad outcomes, with 5-year OS varying from 0% to 25% [161,162].

## 5. Pediatric Ependymomas (pEPNs)

Pediatric ependymomas constitute the third most frequent intracranial tumor. This group emerges in all compartments of the CNS. The most frequent locus is the posterior fossa [163,164]. A slight overall male predominance has been documented [164]. The greatest proportion (90%) of pEPNs arise intracranially. A total of 66% of them are situated in the posterior fossa (PF) and the rest of them are positioned supratentorially [165].

### 5.1. Identity Molecules

Overall, ependymomas originate from radial glial cells and mutations of genes that regulate the proliferation and specialization of neural precursor cells. A copy of the chromosome arm 1q constitutes a predictor of a dismal outcome, especially in the PF ependymomas during childhood [165].

#### 5.1.1. NF-kB Pathway

Canonical NF-kB signaling reacts to tumor necrosis factor-a (TNFa) and interleukin-1 (IL-1) signaling inflammation cytokines. The NF-kB activation relates to RelA- or cRel-containing complexes. The suppression of NF-kB when the inflammation is supposed to cease prolongs the inflammatory response and deters apoptosis [166]. Interestingly, NF-kB constitutes an important inhibitor of pathogen-induced apoptosis of macrophages, at least in vitro. Thereby, NF-kB promotes inflammation via macrophage activation [166] (Figure 20). Initiating genes in the majority of supratentorial ependymomas are fusions between *RELA* of the NF-kB pathway and the gene *C11orf95* caused by a breakage of chromosome 11. This fusion of stem cells can independently promote neoplastic cell proliferation [167].

#### 5.1.2. Hippo Pathway (Figure 21)

The mammalian onco-suppressor Hippo pathway is composed of mammalian sterile-like kinase 1/2 (MST 1/2), Salvador homolog protein 1 (SAV1), large tumor suppressor kinases 1/2 (LATS1/2), monopolar spindle-one-binder protein 1 (MOB1), and yes-associated protein/tafazzin family protein (YAP/TAZ). When the Hippo signaling is active, YAP/TAZ receives phosphate molecules and is either restricted in the cytoplasm or spotted for disintegration. When signaling is inactive, YAP/TAZ enters the nucleus to ignite the transcriptional enhancer factor domain family (TEAD) [168]. Mechanical traits are significant stimuli by which cells apprehend their microenvironment. Hippo’s signaling is promoted by these mechanical stimuli [168]. The frequency of the cells multiplication is inversely proportional to the cell compactness [169]. LATS becomes functional compared to YAP, which remains inert. Furthermore, the overexpression of YAP could reverse the inhibition of growth induced by cell density. Overall, YAP/TAZ is a significant suppressor of Hippo signaling [168].

**Figure 21 cancers-17-01566-f021:**
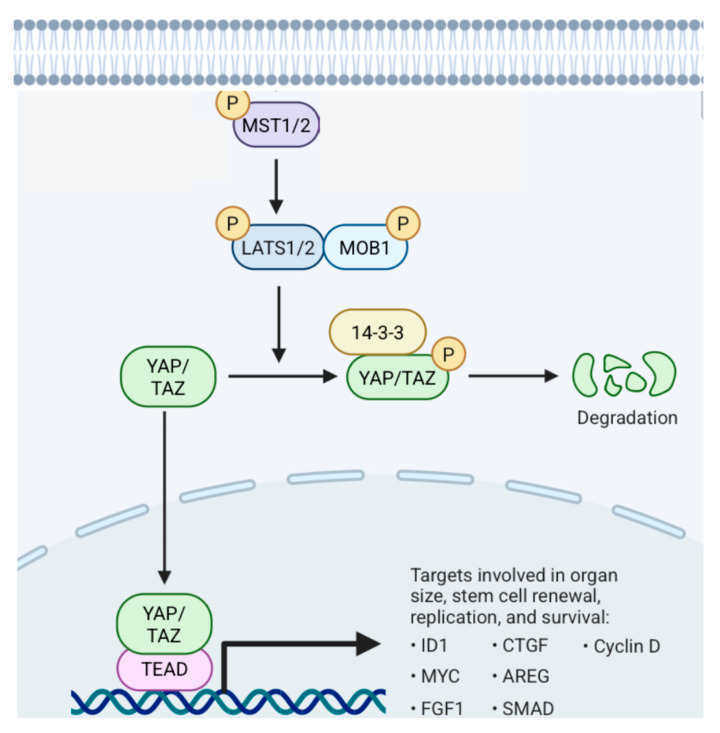
Hippo pathway. MST1/2, SAV1, LATS1/2, YAP, and TAZ are significant components of the signaling. When the Hippo pathway functions, MAP4Ks, MST1/2, and its docking protein, SAV1, add phosphate molecules to LATS1/2 and MOB1A/B [170]. Functional LATS1/2 adds phosphates and suppresses YAP and TAZ, obstructing them from entering the nucleus. MST1 and MST2 are serine/threonine kinases whose activity can be enhanced after their binding with the docking protein SAV1 [168].

For PF EPNs, two distinct molecular subgroups have been consistently identified [171]. These were the PF Group A and Group B, or else ascribed as PFA and PFB [172].

### 5.2. Supratentorial Ependymomas (ST EPNs)

RELA fusions have been solely detected in the supratentorial-RELA subgroup. More than two-thirds of ST EPNs bear a combination of *C11orf95*–RELA fusions [167]. *C11orf95*–RELA fusion proteins enter and incite the canonical NF-κB cascade [165]. These ependymomas manifest a copious capillary network [173]. RelA protein is detected through immunohistochemistry, which is a cost-effective method [174]. The tumor has been traditionally associated with low survival rates [165]. However, recent findings in an exclusively pediatric population have revealed no prognostic disadvantage compared to other subtypes [175].

The other type of ST EPNs is interrelated with amplifications on chromosome 11 at the YAP1 locus [172]. The most well-studied mutation is the YAP1-MAMLD1 fusion. The latter appears exclusively in school-aged children and has an excellent outcome after complete excision, even without adjunct treatment [175,176]. There is another group of supratentorial ependymomas in children that bear the homozygous deletion of *CDKN2A* [172]. In pediatric ependymomas, *CDKN2A* deletion coexists with *Rela* fusion [165]. Lastly, ST EPNs, without the aforementioned molecular alterations, are rare and warrant further observation [177].

### 5.3. Posterior Fossa Ependymomas (PF Ependymomas)

Apart from histological grading in classic and anaplastic ependymomas, prognosis relies on the distance from the midline [165]. Furthermore, malignancies with chromosomal mutations carry an ominous prognosis, especially those with an addition of the q-arm of chromosome 1. Ependymomas of the posterior fossa can be divided into two defined groups [165], which are discussed in detail in the following paragraphs.

PF-A EPNs appear majorly in school-aged children. Boys are usually afflicted with worse survival [13]. PF-A tumors do not harbor alterations at a specific genetic location but rather gains on the long branch of the 1st chromosome in approximately 20% of the cases [171]. Furthermore, excessive additions of methyl groups at CpG-island are present in PF-A tumors [171]. This is attributed to excessive encoding of EZHIP (*CXorf67*) (enhancer of Zeste inhibitory protein) [165]. EZHIP dampens the action of the PRC 2 complex, which suppresses the expression of oncogenes by trimethylating the H3K27 [178]. Diminished trimethylation at the H3K27 location is the hallmark of PF-A EPNs [179]. The deficient methylation pattern is estimated by immunohistochemistry. PF-B ependymoma is found primarily in older children and adults younger than thirty years old. H3K27me3 is detected in normal levels. Girls are afflicted slightly more [165]. They emerge from the obex, and the prognosis is considered good.

### 5.4. Spinal Ependymomas

Spinal ependymoma is infrequent in children. It generally afflicts older patients compared to intracranial malignancies [164]. WHO grade 1 neoplasia is the myxopapillary group. WHO grade 2 tumors are the classic ependymomas. Finally, the anaplastic ependymoma corresponds to WHO grade III tumors [180]. *n-MYC* amplification relates to dismal prognosis [181]. Spinal ependymomas manifest a methylation pattern differentiated from subependymomas, myxopapillary ependymomas, and MYCN-amplified (anaplastic) ependymomas [182].

### 5.5. Current Treatment Status

Maximal safe excision accompanied by focal radiation is the hallmark of therapy. The degree of resection is the most significant predictor of prognosis [147]. We observed 60-month OS after resection rates of 67–80% compared to 22–47% after incomplete excision and irradiation. Unfortunately, gross total resection is possible in 42–62% of the cases [147]. Proton-beam radiation therapy (proton-RT) belongs to the armamentarium of treatment. Patients with local disease demonstrate a 3-year progression-free survival percentage similar to patients who had undergone intensity-modulated RT. An amount of 59.4 Gy is usually applied [183].

To shun the toxicity of radiation in preschooler children, protocols with precedent chemotherapy were introduced. Younger children without metastases and no precedent irradiation had a 60-month OS in 63.4% of the cases [147]. Commencement of RT 10 months after chemotherapy did not bear risks for young children with complete resection. Response to chemotherapy is remarkable, and OS rates are 81.1% [147]. Ependymomas respond well to certain chemotherapeutic regimens [184]. Overall, recurrence portends a dire outcome. The incidence of tumor relapse is moderately less than half of the cases and usually appears within 19 months; some subtypes can relapse 20 years after the initial treatment [185].

## 6. Germ Cell (GC) Tumors

GC neoplasms arise from embryonic cells that have eluded regular neural tube formation. The origin of these tumors is hypothesized to be the result of the deviant translocation of primordial germ cells (PGCs) towards the growing fetal gonadal fold [186]. These tumors reveal overall DNA hypomethylation, and this observation corroborates the former hypothesis [187]. The downregulation of the KIT/KITLG signaling cascade (subtype of tyrosine kinase receptors) caused by BAK1 deletions hinders primordial germ cell death [188] (Figure 22). A gain of 2q and 8q and deficiencies in 5q, 9p/q, 13q, and 15q portend to worse outcomes.

Non-germinomatous germ cell tumors (NGGCTs) manifest neuronal differentiation and increased expression of the Wnt/β-catenin cascade [189]. Chromosome 12p additions are associated with reduced OS and appear in one-third of CNS GCTs and in half of NGGCT cases [190].

KIT/RAS and PI3K/AKT1 alterations link to germinoma formation [168]. MAPK signaling mutations portend thalamic lesions [191]. NGGCT had significantly higher immune cell infiltration, implying an immune-suppression phenotype [192]. Epigenetically, germinoma/seminoma and non-germinoma/non-seminoma are studied distinctively, according to histology and MAPK cascade mutations. Fundamental structural aberrations of sex chromosomes, such as Klinefelter’s syndrome (47, XXY) and Down’s syndrome (Trisomy 21), herald increased chances of CNS GCT [186].

### 6.1. Diagnostic Work-Up

Germ cell tumors produce specific biochemical traits involving alpha-fetoprotein (AFP) and human chorionic gonadotropin (HCG). AFP is secreted, apart from yolk sac, in the liver as well. Embryonal carcinomas also secrete this marker. Levels > 25 ng/mL in serum and/or CSF are regarded as diagnostic, and treatment may commence without biopsy [193].

HCG is produced by the placenta and by non-germinomatous germ cell tumors. Levels more than 50 IU/L in serum and CSF are pathognomonic. Hence, treatment may follow without biopsy [194].

Germinomas exhibit lymphocytic invasion; hence, diagnosis may be laborious due to the scant presence of malignant cells. Therefore, an armamentarium of immunohistochemical markers is used. These involve molecules, such as KIT, OCT3/4, or NANOG (Figure 20). When syncytial cell populations are present, then HCG staining is positive in serum and CSF [193].

Embryonal carcinoma is composed of cells with great nucleoli, organized in niduses with surrounding septa. They present increased mitoses. The immunohistochemical marker is the CD30 molecule. Yolk sac neoplasms are formed by primordial-resembling epithelial cells arranged in a loose network. Immunohistochemical identity includes AFP, along with SALL4 and glypican 3. Choriocarcinomas are markedly identified by cytotrophoblastic tissue and positivity for HCG.

Teratomas also include ectodermal, mesodermal, and endodermal tissue. They may also be classified as mature or immature. Mixed malignant germ cell tumors include the teratoma component and anything of the aforementioned germ cell tumors [194].

When AFP and HCG are negative in both serum and CSF, these molecular markers should be used. OCT3/4 and NANOG are found in germinomas and EC. SOX2 is expressed exclusively in EC; therefore, they can be differentiated. CD30 is also positive in EC [194].

### 6.2. Treatment

In cases of elevated intracranial pressure and clinically stable patients, MRI of the neuraxis should be employed together with serum markers. Cytology of CSF follows during the shunt procedure, which may be combined with biopsy. The endoscopic third ventriculostomy is the preferred procedure. Otherwise, external ventricular drainage may be sufficient, as long as chemotherapy and radiotherapy are very effective, and obstruction may recede [195].

When patients manifest no signs of hydrocephalus, CSF should be extracted as the first step of the diagnosis [196].

Diabetes insipidus is a common comorbidity in the setting of pituitary or hypothalamic infiltration. Hydrocortisone is administered in the perioperative period. Desmopressin should precede surgery [194].

Disease is considered metastatic when ventricles seem dotted intraoperatively or when dissemination is noticed in imaging. Finally, positive immunohistochemistry of ventricular CSF signifies dissemination, even when lumbar CSF is negative [194].

Germinomas are responsive to chemo- and radiotherapy. Traditionally, neuraxis radiotherapy was considered the mainstay of treatment. The accumulation of irradiation and its long-term sequels brought chemotherapy to the foreground [193]. On the other hand, chemotherapy solely cannot attain significant treatment benefits compared to irradiation (50% versus 90%, respectively) [197]. Initially, the radiation dosage was 30.4 Gray (Gy) [198].

Radiation dosage varies from 24 to 40 Gy according to the dissemination of the tumor, where the tumor bed may receive the first fraction followed by a boost dose [199]. Paradoxically, residual lesions do not constitute an inauspicious predictor [200]. Apart from this, approximately one-fifth of the patients with germinoma have metastatic disease. Even in this scenario, irradiation provides thirty-six months of PFS in 95% of the patients [201]. Surgery is discussed in some cases when the lesion’s diameter exceeds two centimeters after neoadjuvant chemotherapy or when teratoma is suspected.

The non-germinomatous GCTs are overall regarded as more malignant than germinomas. Therein, neoadjuvant chemotherapy is administered initially, and then higher doses of radiotherapy follow. The dose varies from 50 to 54 Gy. In cases of teratomas, which are recalcitrant to chemotherapy and irradiation, surgical resection is warranted. Overall, any case should be individualized and discussed [194].

## 7. Future and Experimental Treatments for Pediatric Brain Cancer

Pediatric brain neoplasias present unique challenges for innovative treatment due to the developing nature of children’s brains. Both the severity of the disease and the restrictively permeable blood–brain barrier warrant a special approach. Recent advancements have led to the development of new and experimental treatments that hold promise for improving outcomes in young individuals (Table 3). Novel biological agents, such as monoclonal antibodies, intrathecal or intravenous administration of adoptive cells, cancer vaccines, and cancer cell apoptosis-inducing viruses, are applicable. They target high-grade malignancies, like DIPG and medulloblastomas, or recurrences. Apart from these, cellular pathway inhibitors are used. Thus, aberrant function of these altered proteins is selectively hindered. Finally, non-organic molecules or ultrasound devices modify the permeability of the peritumoral blood–brain barrier, thus enhancing the effect of chemotherapy.

## 8. Conclusions

Based on the current WHO classification, older borderlines have been substantially revised. Tumor behavior and potential cannot be defined exclusively by descriptive histology, since its molecular profile has become a major player. The latter can affect tumor location, safety of resection, and finally the overall prognosis.

A vast spectrum of genetic alterations, including DNA point mutations, fusion of genes’ sequences, and amplifications or deletions of genetic loci, participate in tumorigenesis. DNA methylation status and epigenetic control are of paramount importance. Apart from methylation profiles, histone alterations, microRNAs, and chromatin remodeling complexes contribute to epigenetic control. Tailored diagnostic methods, though, are not available to every institution at any time.

Pediatric CNS malignancies have a different molecular profile compared to adult tumors. The presence of IDH-altered malignancies is uncommon. Histone alterations and NTRK mutations predominate in infancy and childhood. The first pattern is associated with high-grade lesions. On the contrary, the second one responds to both low- and high-grade lesions. Based on the methylation profile, they do not constitute, though, a distinct subtype. Furthermore, pediatric high-grade gliomas do not derive from low-grade lesions, which is the rule in the adult population. As far as pLGGs are concerned, detection of *BRAF* mutations renders these tumors liable to novel treatment approaches, such as BRAF inhibitors.

Embryonal tumors, despite their common morphology under a microscope, exhibit molecular diversion. This further corresponds to significant variation in the prognosis, even within the subgroups. Epigenetic dysregulation is also present herein.

Ependymomas are now studied as separate groups, mainly according to their location. This is remarkable in PF EPNs, whose occurrence, surgical outcomes, and clinical course are completely different. They share common epigenetic dysregulation patterns and anatomical locations with DIPGs.

Treating physicians, including neurosurgeons, should be familiar with the basic aspects of the WHO classification, since the therapeutic strategy has become multimodal.

In summary, the current WHO classification that includes complex molecular features for pediatric neoplasms tends to define more explicitly the distinction of malignant from benign tumors, providing clinicians with time-saving diagnoses and better knowledge for prognosis and overall survival for each group. In addition, it paves the way for the development of new and targeted therapeutic strategies that are less toxic and less harmful for the patient compared to traditional strategies.

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
