# Peer review of "The Molecular Basis of Pediatric Brain Tumors: A Review with Clinical Implications"

_cancers, 2025, doi:10.3390/cancers17091566_

Round 1
Reviewer 1 Report
Comments and Suggestions for Authors
Review Report
Title: “The molecular basis of pediatric brain tumors. An update with clinical implications”
Summary
This narrative review provides a comprehensive overview of the molecular mechanisms underlying pediatric brain tumors. The authors discuss key signaling pathways — including MAPK, PI3K-Akt-mTOR, and Wnt — as well as epigenetic alterations and cell cycle dysregulation. The manuscript further delineates the molecular profiles of various tumor subtypes, such as low-grade gliomas, high-grade gliomas, and embryonal tumors. It attempts to correlate these findings with current therapeutic strategies and prognostic implications.
Major Strengths
Comprehensive Coverage: The review successfully encompasses a wide range of molecular mechanisms and genetic alterations, which is commendable given the complexity of pediatric brain tumors. Including multiple signaling pathways (e.g., tyrosine kinase receptors, MAPK, PI3K-Akt-mTOR, epigenetic modulators) provides a valuable resource for clinicians and researchers alike.
Detailed Discussion on Tumor Subtypes: The document thoroughly explores grade and high grade gliomas along, with tumors by pointing out the variations in genetic changes and how they manifest clinically. This thorough examination helps shed light on the nature of brain tumors, in children.
Integration of Clinical Implications: The attempt to correlate molecular findings with clinical outcomes and treatment strategies is a strong point. This integrated approach helps bridge the gap between bench research and bedside applications, underscoring the importance of molecular profiling in tailoring treatment.
Major Weaknesses and Areas for Improvement
Organizational Clarity and Structure: While the manuscript is rich in content, its organization can be improved. The narrative becomes overly detailed in several sections, sometimes overwhelming the reader. A more apparent segmentation - perhaps by summarizing key points at the end of each significant section - would improve readability and facilitate comprehension.
Redundancy and Repetition: Some parts explain the processes repeatedly, so condensing the discussion can improve the manuscript's clarity and prevent essential discoveries from getting buried in too much detail.
Visual Aids and Figure Quality: The figures provided (such as pathway diagrams) are helpful. Might need some improvement in terms of clarity and labeling to be more effective, for readers especially when explaining signaling pathways in the text content. Therefore, I recommend redrawing all the figures using editing tools or considering professional editor services to guarantee consistent visuals.
Minor Points
Consistency in Nomenclature: Maintaining uniformity in naming conventions is essential. Make sure that you keep the gene and protein names consistent throughout your paper in terms of capitalization and formatting to ensure clarity in a review.
Reference Integration: While the manuscript has references overall, enhancing the integration of these references into the discussion by connecting them to clinical outcomes or comparing various studies would bolster the points being made.
Conclusion Section: In wrapping up, it's important to outline the takeaways and clearly outline how these findings could impact real-world clinical settings moving forward. A well-crafted ending should ensure that readers grasp the significance of the study's findings and how they may shape future investigations in the field.
Recommendation
The manuscript provides an insightful overview of the foundations of pediatric brain tumors as they stand now; however, substantial revisions are needed before publication to tackle problems surrounding structure, repetition, and coherence. Improving the integration of implications and enhancing aids will be vital to guarantee that the review is informative and easily understood by its intended readership.
Author Response
Dear Reviewer,
We are grateful for your productive recommendations which enabled us to revise significantly our manuscript. We remain at your disposal for any further recommendations.
We improved the quality of the figures. Repetition of phrases has also been evaluated and corrected. Furthermore, an abbreviation chart has been added in our aim to have a consistency in nomenclature. We have added a section concerning novel treatments. Finally, we altered the Conclusions paragraph stressing the utility of WHO Classification in the diagnostic and therapeutic field.
Reviewer 2 Report
Comments and Suggestions for Authors
The manuscript "The Molecular Basis of Pediatric Brain Tumors: An Update with Clinical Implications" presents a comprehensive review of the genetic and molecular mechanisms underlying pediatric brain tumors. The authors provide an in-depth discussion of various signaling pathways, genetic mutations, and clinical implications of these malignancies. The paper is well-structured and informative but requires substantial revisions to enhance clarity, coherence, and scientific rigor. Including more structured tables, more precise definitions of abbreviations, and a more targeted discussion on clinical applications will significantly enhance the manuscript’s quality.
Improvement Suggestions:
- The abstract presents a broad overview of molecular mechanisms but lacks clarity in the distinction between different tumor types. The phrase "These lesions are the result of the aberrant cell signaling steps" (Line 31) is vague. Additionally, the statement "It can occur either due to histones mutations or to inappropriate binding or unbinding of chromatin on histones" (Lines 33-34) requires clarification regarding how these epigenetic changes contribute to tumorigenesis.
- The introduction mentions that "Childhood brain tumors are the commonest solid malignancy and second among all pediatric neoplasias following leukemia" (Lines 56-57). This is repeated within the same paragraph. The redundancy should be removed.
- The definitions of carcinogenesis stages (initiation, promotion, and progression) are helpful, but some descriptions are unclear. For example, "The promotion phase entails the distinct proliferation of the initiated cell to shape a class of preneoplastic cells" (Line 75) could be expanded to explain the molecular basis of tumor promotion in pediatric brain tumors.
- The manuscript introduces multiple abbreviations (RTK, TKD, MAPK, PIP3, etc.) without defining them upon first use. For example, "RTKs are highly-specified cell surface receptors for polypeptide growth factors, hormones, and cytokines" (Line 94) should include the full form of RTK in parentheses at its first mention.
- The discussion on FGFR1 alterations (Lines 104-110) lacks a systematic approach. Some mutations are mentioned with percentages, while others are only described qualitatively. A table summarizing key mutations and their frequencies in different tumor types would improve clarity.
- The explanation of Figure 4 (Lines 145-147) is difficult to follow. The description should explicitly state how each step in the pathway leads to oncogenesis rather than simply listing the components. Furthermore, the connection between NF1 deficiency and excessive pathway activation should be elaborated with a specific clinical example.
- While MYB/MYBL1 alterations are discussed, they are not compared to other genetic changes in gliomas.
- The description of pilocytic astrocytomas (Lines 191-234) is overly technical and lacks coherence in explaining the clinical significance of histopathological features. The histological subtypes should be clearly distinguished with corresponding implications for prognosis and treatment.
- The discussion on epigenetic regulation (Lines 304-328) is conceptually critical but lacks specific examples. For instance, mentioning how H3K27M mutations are linked to prognosis in DIPGs would provide clinical relevance.
- The categorization of high-grade gliomas (Lines 307-319) should be more intuitive. Rather than listing molecular alterations without clear differentiation, a hierarchical structure first categorized by WHO grade and then by molecular subtype would improve clarity.
- The manuscript extensively explains the WNT and SHH pathways (Lines 604-638), but there is redundancy in defining transcriptional activation mechanisms.
- The treatment discussion is too focused on traditional therapies (surgery, chemotherapy, radiation) without addressing emerging targeted therapies (Lines 564-582). Incorporating data on BRAF and EZH2 inhibitors would enhance clinical applicability.
- The conclusion should more explicitly summarize the clinical implications of the molecular classifications discussed in the review. Currently, it lacks emphasis on how these classifications can be translated into improved diagnostic and therapeutic strategies.
Author Response
Dear Reviewer 2 we are grateful for your recommendations and we have proceeded to the necessary changes. Your comments helped us significantly to change our manuscript. We highlighted the changes in the text and we provide you with point-to-point responses in your recommendations. We remain at your disposal for any further recommendations.
- The abstract presents a broad overview of molecular mechanisms but lacks clarity in the distinction between different tumor types. The phrase "These lesions are the result of the aberrant cell signaling steps" (Line 31) is vague. Additionally, the statement "It can occur either due to histones mutations or to inappropriate binding or unbinding of chromatin on histones" (Lines 33-34) requires clarification regarding how these epigenetic changes contribute to tumorigenesis.
Response: Line 31. We clarify how mutated signalling proteins result in aberrant proliferation and tumorigenesis.
Line 34. We justified the way epigenetic loss of control contributes in tumorigenesis.
- The introduction mentions that "Childhood brain tumors are the commonest solid malignancy and second among all pediatric neoplasias following leukemia" (Lines 56-57). This is repeated within the same paragraph. The redundancy should be removed.
Response: We have deleted the sentence of line 56, which is also mentioned in line 52.
- The definitions of carcinogenesis stages (initiation, promotion, and progression) are helpful, but some descriptions are unclear. For example, "The promotion phase entails the distinct proliferation of the initiated cell to shape a class of preneoplastic cells" (Line 75) could be expanded to explain the molecular basis of tumor promotion in pediatric brain tumors.
Response: We explain how specific genetic alterations may elongate this phase in paediatric brain lesions.
- The manuscript introduces multiple abbreviations (RTK, TKD, MAPK, PIP3, etc.) without defining them upon first use. For example, "RTKs are highly-specified cell surface receptors for polypeptide growth factors, hormones, and cytokines" (Line 94) should include the full form of RTK in parentheses at its first mention.
Response: We have proceeded to the requested changes. We also added an abbreviations list after Conclusions aiming to maintain consistency in nomenclature.
- The discussion on FGFR1 alterations (Lines 104-110) lacks a systematic approach. Some mutations are mentioned with percentages, while others are only described qualitatively. A table summarizing key mutations and their frequencies in different tumor types would improve clarity.
Response: We have added a new table in our attempt to systematically demonstrate the classification of FGFR alterations.
- The explanation of Figure 4 (Lines 145-147) is difficult to follow. The description should explicitly state how each step in the pathway leads to oncogenesis rather than simply listing the components. Furthermore, the connection between NF1 deficiency and excessive pathway activation should be elaborated with a specific clinical example.
Resonse: We have added in the legend of the image the specific function of each protein.Therein, the impact of deficient NF1 function is discussed. We have also a clinical case of an optic glioma in a patient with neurofibromatosis, which is characteristic of NF1 deficient function.
- While MYB/MYBL1 alterations are discussed, they are not compared to other genetic changes in gliomas.
Response: In Section 3.1.1.4 we added a paragraph at the end of this subsection comparing MYB/MYBL1 gliomas with the other groups (both their frequency and prognosis).
- The description of pilocytic astrocytomas (Lines 191-234) is overly technical and lacks coherence in explaining the clinical significance of histopathological features. The histological subtypes should be clearly distinguished with corresponding implications for prognosis and treatment.
Response: In Section 3.1.2.5 we reduced the use of so many technical terms and altered the structure of the paragraph. Now, the molecular subtypes are correlated with their respective treatment and prognosis.
- The discussion on epigenetic regulation (Lines 304-328) is conceptually critical but lacks specific examples. For instance, mentioning how H3K27M mutations are linked to prognosis in DIPGs would provide clinical relevance.
Response: In 3.2.1.1 and 3.2.1.2 we explain the way H3K27 mutations contribute in epigenetic dysregulation and tumorigenesis in hindbrain area.
- The categorization of high-grade gliomas (Lines 307-319) should be more intuitive. Rather than listing molecular alterations without clear differentiation, a hierarchical structure first categorized by WHO grade and then by molecular subtype would improve clarity.
Response: In subsection 3.2 we clarify the classification by adding an extra Table concerning the categorization of High-Grade Gliomas.
- The manuscript extensively explains the WNT and SHH pathways (Lines 604-638), but there is redundancy in defining transcriptional activation mechanisms.
Response: In Section 4 transcriptional activation mechanisms for both Wnt and SHH pathways are now referred at the legends of the respective figures. We have not changed the images, but we refer the pertinent transcription factors.
- The treatment discussion is too focused on traditional therapies (surgery, chemotherapy, radiation) without addressing emerging targeted therapies (Lines 564-582). Incorporating data on BRAF and EZH2 inhibitors would enhance clinical applicability.
Response: A separate segment, which includes a Table, as well, has been added before the Conclusions section.
- The conclusion should more explicitly summarize the clinical implications of the molecular classifications discussed in the review. Currently, it lacks emphasis on how these classifications can be translated into improved diagnostic and therapeutic strategies.
Response: We added a paragraph for the utility of WHO Classification in the diagnostic and therapeutic fields. We also altered significantly the structure of this paragraph.
Reviewer 3 Report
Comments and Suggestions for Authors
Antoniades et. al. present a review focusing on the molecular basis of pediatric brain tumors. The review covers all the major pediatric brain tumor types in detail. However this review was several flaws that need to be addressed prior to publication.
Comments:
- The figure quality is not up to publication quality. Authors should use software similar to Biorender to make professional figures for this review
- Authors should consider an overarching figure (flowchart) or table that shows all the major brain tumor types and then the mutations/molecular alterations that are key. Currently, it isn't easy to put this tumor into a working framework for clinicians as each section is disjoint from the next. A summary figure or table would be very helpful for this.
- It is not clear what this review adds to the current literature. The molecular profile of pediatric brain tumors has been reviewed in several existing publications over the last 5 years. The authors need to include some novelty/interesting new data/new framework to think about peds brain tumors in this review to differentiate from existing reviews.
There are several English grammar errors and syntax problems throughout the manuscripts. Authors should work to remedy this before publication. I would recommend using an editing service.
Author Response
Dear Reviewer 3
We deeply tank you for the time you spent to review our manuscript. Your comments helped us to improve our manuscript. We remain at your disposal for further recommendations.
Response:
We have added flowcharts for each tumour group and have improved the quality of the figures using Biorender. Apart from this, an additional section along with a Table related to novel treatments is now included. We intended to render molecular interactions comprehensible to clinicians other than oncologists or pathologists.
If Editor considers linguistic improvement we would gladly proceed to it before publication.
Reviewer 4 Report
Comments and Suggestions for Authors
There are some typographical errors.
Human chorionic gonadotropin should be HCG, not HGG.
The treatment strategies for each tumor are too simple. Those descriptions did not provide useful information for clinicians.
Author Response
Dear Reviewer,
We really appreciate the time you spent for evaluating our manuscript. We remain at your disposal for any further recommendations.
Response:
We have corrected the spelling mistakes. We also added a new section and a table related to novel treatments. We hope that this could provide some useful information in the manuscript.
Round 2
Reviewer 2 Report
Comments and Suggestions for Authors
The improvements made by the authors are convincing to accept the manuscript in the present form.
Author Response
Dear Reviewer 2
We appreciate your comments and we would like to thank you for giving us the opportunity to revise our manuscript.
Reviewer 4 Report
Comments and Suggestions for Authors
I recognized significant improvements in this manuscript.
Author Response
Dear Reviewer
We would like to thank you one more time for your recommendations.